# Important Roles and Potential Uses of Natural and Synthetic Antimicrobial Peptides (AMPs) in Oral Diseases: Cavity, Periodontal Disease, and Thrush

**DOI:** 10.3390/jfb13040175

**Published:** 2022-10-03

**Authors:** Albert Donald Luong, Alyah Buzid, John H. T. Luong

**Affiliations:** 1Department of Epidemiology and Environmental Health, School of Public Health and Health Professions, University of Buffalo, Buffalo, NY 14215, USA; 2Department of Chemistry, College of Science, King Faisal University, P.O. Box 380, Al-Ahsa 31982, Saudi Arabia; 3School of Chemistry and Analytical & Biological Chemistry Research Facility (ABCRF), University College Cork, College Road, T12 YN60 Cork, Ireland

**Keywords:** antimicrobial peptides, antifungal, natural peptides, synthetic peptides, dental caries, oral cavity

## Abstract

Numerous epithelial cells and sometimes leukocytes release AMPs as their first line of defense. AMPs encompass cationic histatins, defensins, and cathelicidin to encounter oral pathogens with minimal resistance. However, their concentrations are significantly below the effective levels and AMPs are unstable under physiological conditions due to proteolysis, acid hydrolysis, and salt effects. In parallel to a search for more effective AMPs from natural sources, considerable efforts have focused on synthetic stable and low-cytotoxicy AMPs with significant activities against microorganisms. Using natural AMP templates, various attempts have been used to synthesize sAMPs with different charges, hydrophobicity, chain length, amino acid sequence, and amphipathicity. Thus far, sAMPs have been designed to target *Streptococcus mutans* and other common oral pathogens. Apart from sAMPs with antifungal activities against *Candida albicans*, future endeavors should focus on sAMPs with capabilities to promote remineralization and antibacterial adhesion. Delivery systems using nanomaterials and biomolecules are promising to stabilize, reduce cytotoxicity, and improve the antimicrobial activities of AMPs against oral pathogens. Nanostructured AMPs will soon become a viable alternative to antibiotics due to their antimicrobial mechanisms, broad-spectrum antimicrobial activity, low drug residue, and ease of synthesis and modification.

## 1. Introduction

Tooth decay or dental caries dates back to several thousand years BC [1] as a cause of tooth loss and tooth root breakdown. Besides viruses and protozoa, there are over 500–700 different microbes in the human mouth [2] including *Candida albicans* [3]. Oral microbes are anchored on dental surfaces to form dental plaque, a sticky film on teeth. They metabolize sugars into organic acids, resulting in enhanced demineralization of tooth surfaces [4,5,6] and damaging tooth structures including enamel, dentin, and cementum. *Streptococcus mutans*, *S. sobrinus*, *Actinomyces* spp., and *Lactobacilli* have been identified as primary cariogenic bacteria [7], whereas *Actinobacillus actinomycetemcomitans* is attributed to periodontal disease, especially in young people. Nonetheless, the biggest culprit of starting cavities is still *S. mutans*, a facultatively anaerobic and Gram-positive coccus. This bacterium in high numbers metabolizes sucrose to large quantities of extracellular glucan, a glucose polymer, to colonize tooth hard surfaces. It also transports and metabolizes different sugars into organic acids (acidogenicity) and flourishes at low pH (aciduricity) [8]. The classification of *S. mutans* into eight different serological groups [9] or possibly only four [10] is related to the linkages and compositions of its cell wall polymers (rhamnose-glucose polysaccharides. Together with *S. mutans*, *L. acidophilus*, and *A. viscosus*, a consortium of different bacteria colonizes and forms pathogenic plaque biofilms, a major contributor to oral infectious diseases. Tooth decay can also be associated with *S. penguins*, *S. mitis*, and *S. milleri. Enterococcus faecalis*, which are often found in teeth with pulp necrosis. Oral pathogens are also classified as facultative anaerobic bacteria (*Lactobacillus* ssp., *Streptococcus* ssp., *Actinomyces* ssp., *Veillonella* ssp., and *Porphyromonas* ssp., obligate anaerobic bacteria (*Fusobacterium ssp*. and *Tannerella* ssp.) and other common aerobic bacteria (*Pseudomonas* ssp. and *Staphylococcus* ssp. Periodontal infections are caused by a consortium of two anaerobes (*P. gingivalis* and *Fusobacterium nucleatum*), one obligate anaerobe (*Prevotella intermedia*), and one facultative anaerobe (*Aggregatibacter actinomycetemcomitans*). Beyond bacterial infections, candidiasis or oral thrush is a condition in which *Candida albicans*, a normal yeast, accumulates on the mouth lining to cause creamy white lesions. Chronic periodontitis is caused by overpopulated anaerobes: *P. gingivalis*, *Tannerella forsythia*_,_ and *Treponema denticola.*

In ancient China [11], arsenic trioxide was used to alleviate tooth pain and this inorganic compound is also used to treat acute promyelocytic leukemia. Under a microscope, Antonie van Leeuwenhoek, commonly known as one of the first microscopist and microbiologist, observed the bacterial presence in his plaque [12]. The metabolism of sugars to acids was then proposed by Miller, leading to the demineralization of teeth [13]. *S. mutans* isolated by Clarke was identified as a bacterium responsible for dental caries [14]. Based on a hamster model, Keyes [15] confirmed the bacterial presence in dental caries and the elapsed time is an important factor in caries etiology [16]. After such events, antibiotics or other antimicrobial compounds have been prescribed to treat or prevent oral diseases [17,18,19,20,21]. The reduction in bacterial load and antibiofilm formation can be effectuated by fluoride, chlorhexidine, phenol, and triclosan [22,23]. However, fluoride and triclosan have some safety concerns, whereas the extended use of chlorhexidine decolorizes soft and hard tissues [24] and triggers hypersensitivity. Phenolics often cause irritation and burning sensations [25]. Anionic peptides are not addressed in this review, albeit negatively charged peptides have been isolated from microorganisms with antibacterial activity [26]. The use of small compounds in oral therapeutics is summarized in Table 1.

This review focuses on native AMPs as alternative therapeutics for oral infections, mainly dental cavities, thrush, and periodontitis. Cationic AMPs exhibit a broad antimicrobial spectrum with fast action as they display electrostatic interactions with bacterial membranes with an opposite charge. Some are α-helical after their interaction with anionic phospholipid membranes or structure-promoting solvents, e.g., trifluoroethanol. Other AMPs are the β-sheet due to the formation of intramolecular disulfide bonds from their cysteine residues. Some AMPs can have uncommon AA such as tryptophan, histidine, or proline residues. The deployment of novel drug delivery systems for AMPs is also not included in this review. This review will cover the current use of natural AMPs and the development of their synthetic counterparts for targeting oral pathogens. Synthetic AMPs are designed to overcome some serious drawbacks of natural counterparts such as high cytotoxicity, instability, and in vivo short half-life. In brief, seven common oral diseases are dental cavities; gingivitis; periodontal disease; thrush (an overgrowth of *Candida albicans* fungus leads to this infection (most common in people diagnosed with HIV); hand, foot, and mouth disease (caused by the virus known as Coxsackie A16); herpangina (related to hand, foot, and mouth disease); and canker sores (lesions in the mouth and gum tissues). The precise cause of canker sores remains unclear, though this oral disease may be caused by a combination of factors including *Helicobacter pylori*, the same bacterium that causes peptic ulcers [42]. Viral infection for oral diseases is beyond the scope of this review; however, the subject of viral infections of the oral cavity is available in the literature [43] with 51 cited papers. The potential use of peptides to combat viral infectious diseases is also covered by Al-Azzam et al. [44] with 235 cited papers. 

## 2. Treatment by Antibiotics

Common antibiotics used for treating cavities include penicillin, tetracycline, metronidazole, macrolides, and clindamycin; however, the antibiotic resistance of oral bacteria was not thoroughly investigated [45,46]. Thus, the use of systemic antibiotics for oral diseases has gradually diminished, except for the prevention of dental infections after surgical interventions, despite the biofilm resistance to amoxicillin and metronidazole [47]. These two antibiotics or co-amoxiclav (amoxicillin mixed with clavulanic acid) are often prescribed, whereas clindamycin is intended for penicillin-allergic patients [48]. The killing effect, side effects, and drug resistance of these antibiotics are summarized in Table 2. Antibiotics are not used to treat candidiasis (*Candida albicans*), which requires antifungal medications such as caspofungin, micafungin, anidulafungin, fluconazole, and amphotericin B. Different antibiotics are prescribed to target anaerobic bacteria, specifically *Streptococci* by metronidazole, clindamycin for aerobic and anaerobic pathogens, ciprofloxacin and tetracycline for G^+^ (Gram-positive) and (G^−^) Gram-negative bacteria, ampicillin for G^+^ bacilli, mostly against aerobic bacteria and penicillin for G^−^ bacilli. Three macrolides, erythromycin, azithromycin, and clarithromycin, are prescribed to target *S. mutans*. 

## 3. The Efficacy of Natural Antimicrobial Peptides (AMPs)

AMPs can have 10–24 amino acids, AA (short), 25–50 AA (medium), and 50–100 AA (long) with corresponding molecular weights of 2–20 kDa. They are in the epithelial lining, blood, and lymphatic tissues and serve as one of the first defenses against pathogens. AMPs can be neutral or charged (cationic vs. anionic); however, common AMPs with Lys and Arg (or sometimes His) residues [66] carry a net-positive charge (+2 to +13) (Table 3). Only 10% of histidine is protonated as its side chain has a pKa of ~6.5. Glycine (2-aminoethanoic acid) and alanine (2-aminopropanoic acid) are the two simplest AAs.

Unlike proteins, AMPs usually have a few specific amino acids, e.g., proline in apidaecin (GNNRPVYIPQPRPPHPR**L**18-AA, proline-rich peptide) and pyrrhocoricin with 20 AAs (VDKGSYLPRPTPPRPIYNRN from the firebug *Pyrrhocoris apterus*); His-5 with Try, Arg, Gly, and His; or nisin with modified AAs [67]. AMPs are categorized as linear α-helical peptides (human cathelicidin LL-37-LLGDFFRKSKEKIGKEFKRIVQRIKDFLRNLVPRTES) and histatin-5 in trifluoroethanol), β-sheet peptides (gomesin, ZCRRLCYKQRCVTYCRGR, 18-AAs, a Cys-rich peptide released by hemocytes of *Acanthoscurria gomesiana*), linear extension structure (indolicidin with 13 AAs isolated from neutrophil blood cells of cows, ILPWKWPWWPWRR-NH_2_), and both α-helix and β-sheet peptides (α1-purothionin with 45 AA residues with 8 cysteines, a wheatgerm cysteine-rich lipid-binding protein and lytic toxin). The commonly accepted mechanism of AMPs is attributed to their capability to damage bacterial membranes. AMPs with positive charges from their positive AA residues bind to G^−^ (lipopolysaccharides in their outer membrane) and G^+^ bacteria (teichuronic and teichoic acids in cell walls) with negative charges via electrostatic interactions. The binding event is governed by the AA composition and sequence of AMPs and bacterial biomolecules located in their membrane with an opposite charge, e.g., anionic phospholipids. Consequently, most AMPs permeabilize the bacterial membrane to cause significant damage or small defects, which in turn dissipate the transmembrane potential, leading to cell death. Pore and non-pore models are generally accepted as the mechanism of antimicrobial action. Two models have been suggested: barrel stave pore and toroidal pore. Several models for the non pore theory have been proposed, including the carpet model, the detergent model, the molecular shape model, etc. The carpet model is the most-cited model, which demonstrates parallel deposition of AMPs on the cell membrane, causing bilayer destabilization. In all models, AMPs need to reach a certain threshold concentration in the cell membrane prior to disruption. These peptides targeting the cell membrane are suited for application in dental or medical treatment. The mechanisms of antimicrobial action were discussed by Wang et al. [68] concerning the potential use of antibiofilm peptides against oral biofilms.

For a given peptide/lipid ratio and affinity binding, amphipathic AMPs can reorient themselves to interact favorably with bacterial membranes [69]. For the pore models, the barrel-stave model involves the assembly of AMP-lined transmembrane pores as oriented peptides are inserted into the membrane, enabling peptide–peptide interactions (Figure 1 and Box 1). This causes the intracellular contents of bacteria to leak out, resulting in cell lysis. Aggregated AMPs cause a continual bending of the lipid monolayer through the pores, according to the toroidal model. AMPs can translocate into the cytoplasm and interact with specific biomolecules to inhibit bacterial growth and replication. Hydrophobicity dictates the partitioning extent of AMPs in the lipid bilayer, a prerequisite for membrane permeabilization. The amphipathic structure is critical for AMPs to penetrate the membrane to form hydrophobic channels or pores. Amphipathic AMPs attack the membrane by interacting with its hydrophobic–lipids. Another important parameter is the hydrophobic moment, which controls the switching of AMPs from a polar face to a nonpolar face. Some peptides with all-L or all-D AA exhibit different antimicrobial activities, e.g., apidaecin (18 to 20 Aas, Pro-rich, GNNRPVYIPQPRPPHPRL) and drosocin (19 AAs, Lys and Arg-rich, GKPRPYSPRPTSHPRPIRV) [69]. Accordingly, some AMPs might bind to bacteria via specific bacterial receptors. For none pore models, the carpet model deciphers the AMP adsorption on an entire bacterial membrane surface, resulting in membrane disruption and the formation of micelles.

Box 1The antimicrobial mechanisms of AMPs.Step 1: Cationic AMPs bind to the negatively charged surfaces of Gram-negative (outer membrane) or Gram-positive (cell wall) bacteria.Step 2: AMPs then accumulate on the bacterial membrane surface and adopt their stable secondary structure.Step 3: With increasing peptide-lipid ratio on the bacterial membrane, the AMP hydrophobic region gradually interacts with phospholipid heads on the bacterial membrane.Step 4: When AMPs reach a threshold concentration, they would disrupt the bacterial membrane, causing cell lysis. However, AMPs may also act intracellularly, including the inhibition of DNA/RNA or protein synthesis.

AMPs can also inhibit bacterial cell-wall formation, breaking down DNA or RNA in bacterial plasma, and causing protein defragmentation or degradation. These peptides also induce an autolysin effect and inhibit bacterial enzyme activity [72]. Some AMPs in the cathelicidin family secreted by host cells play a pivotal role in oral wound healing by inhibiting osteoclastogenesis [44]. Cathelicidin-related peptides, e.g., LL-37, prevent alveolar bone destruction in periodontitis by inhibiting calcineurin activity and nuclear translocation of T-cells [73]. 

Inducible AMPs are produced to counteract invading pathogens including cariogenic bacteria. At least a few thousand AMPs were identified from animals, bacteria, fungi, and plants. Human oral AMPs are released to defend against pathogens, repair cellular tissues, and suppress oral cavities. However, natural AMPs have low bioavailability as they are vulnerable to proteases and unable to eradicate *S. mutans*, *C. albicans*, and other oral pathogens. Nevertheless, they balance the oral microflora and control the overgrowth of harmful pathogens. Three native AMPs, histatins, defensins, and cathelicidins, exhibit a wide antimicrobial spectrum against oral G^−^/G^+^ bacteria, yeasts, and viruses. The release of reactive oxygen species (ROS) by His-5 is considered the principal mode for killing *C. albicans*. Both His-3 and His-5 can bind to metals as discussed earlier and the release of ROS damages cell membranes, DNA, and other biomolecules [74].

### 3.1. Histatins 

Parotid and submandibular salivary duct cells [75] synthesize histidine-rich histatins at ~50–425 µg/mL in healthy adults [76]. Histatins (His-1, His-3, and His-5) are up to 85% of the salivary histatins and His-5 has the most potent fungicidal activity [77]. They bind to metal ions and regulate oral hemostasis [78]. Histatins are also attributed to attained enamel pellicles [79] due to their strong affinity for enamel surfaces. Histatin-1 (His-1) is protected from proteolysis and dental demineralization as it is absorbed by hydroxyapatite crystals. Histidine is crucial to the function of histatins as the removal of this amino acid (especially in His-5) results in reduced antifungal activities against common yeasts such as *C. glabrata*, *C. krusei*, *Cryptococcus neoformans*, and *Saccharomyces cerevisiae* by its disruption of the fungal plasma membranes. The reactive N-terminals in His-3 and His-5 bind to metals, especially copper and nickel [80] to generate the ROS, which in turn damage the membranes of cell organelles and DNA, leading to fungal and bacterial cell death [80]. Based on the AA sequences of 12 histatins, His-2 is derived from His-1 (Table 4). The remaining histatins are the proteolytic products of His-3. 

### 3.2. Defensins

Defensins (α-defensin, hNP, and β-defensin, hBD) are short peptides (Mw = 4–5 kDa) with a β-sheet structure due to the formation of 3–4 intramolecular disulfide bonds by 6–8 Cys residues. Based on the Cys spacing and the connecting pattern of the three disulfide bonds, defensins are classified as α or β. They are cationic because their AAs consist of the Arg and Lys residues. Besides fungi and herpes simplex, defensins can eradicate G^+^ and G^−^ bacteria [81,82,83] and enhance antibiotic efficacy. In G^-^ bacteria, the target is lipopolysaccharide (LPS), whereas teichoic acid is the target in G^+^ counterparts. The defensins also target membrane-rich phospholipids, common to both G^+^ and G^-^ bacteria. Among the four mature human peptides, hNP-1, hNP-2, and hNP-3 exhibit a very similar AA sequence [84]. hNP-2 has 29 AAs, compared with 30 AAs for hNP-1 and hNP-3. hNP-1,2, and 3 are most abundant in the saliva (~99%), whereas the hNP-4 level is 100-fold lower [85]. hNP-1 is more effective than hNP-3 against two G^-^ bacteria (*E. coli* and *P. aeruginosa*) and G^+^
*S. aureus* [85]. Antimicrobial activities against (G^−^) *E. coli*, (G^+^) *S. faecalis,* and *C. albicans* [86] are noted for hNP-4 with 33 AAs. However, the oral cavity lacks hNP-5 and hNP-6 [87]. 

The *β*-defensin family of epithelial cells has six members (hBD-1–6) [88], whereas the oral cavity has only three defensins, known as hBD-1, hBD-2, and hBD-3 [89], designated as hBD 1,2,3. *β*-defensins are amphipathic as their structure consists of both hydrophilic and hydrophobic domains [90]. The salivary levels of hBD-1 and hBD-2 are very similar (39 ng/mL versus 33 ng/mL) [91], compared to 0.31 μg/mL for hBD-3 [92]. The microbial activities of hBD against some oral pathogens are summarized in Table 5. 

A mechanism for the killing effect of defensins has been proposed [99], involving their binding with anion lipids of the bacterial membrane. The binding event initiates the formation of multimeric pores, membrane permeation, and intracellular biomolecule leakage. Besides its defense against pathogens, the β-defensin family has other functions in cell proliferation, wound healing, and cancer [100]. The AA sequences of *α*- and *β*-defensins are illustrated in Table 6 together with the target pathogens. 

### 3.3. Human Cathelicidin (LL-37)

Among 30 identified members, LL-37 with 37 AAs is the only amphipathic *α*-helical cathelicidin expressed in humans. This peptide is also known as CAMP, hCAP-18 (MW = 18 kDa), or FALL-39, which is derived from hCAP-18, an inactive form in cytoplasmic granules and lamellar bodies by proteolysis of kallikreins or proteinase. Over 50% of LL-37 AAs and two leucines at the N-terminus are hydrophilic. With an *α*-helix structure, several AA of LL-37 form intramolecular hydrogen bonds and its antibacterial activity is related to *α*-helicity [103]. Salivary LL-37 is about 0.14–3 μg/mL, well below the inhibiting concentration required for *P. gingivalis* (MIC = 125 μg/mL) and *A. actinomycetemcomitans* (30–60 μg/mL) [104]. LL-37 binds to the bacterial LPS [105] to cover the bacterial membrane in a carpet-like manner as described earlier (Figure 1). Its mechanism mode is similar to that of detergent to form micelles [106] and disrupt the anaerobic biofilm. LL-37 is used to treat dental caries and chronic periodontal diseases [107,108]. LL-37 has moderate antimicrobial activities against *Enterococcus genera, Staphylococcus*, *Pseudomonas*, and *Escherichia* [108]. Increasing LL-37 levels promote rapid wound healing and protect against lesion infection. Salivary levels of LL-37 are correlated with manifested mucosa lesions in oral lichen planus (chronic oral inflammation) patients [109]. After its passage from the outer membrane of G^-^ bacteria, LL-37 interacts with the peptidoglycan layer and the inner cytoplasmic membrane [110]. 

## 4. Synthetic Antimicrobial Peptides (sAMPs) 

The conquest for synthetic antimicrobial peptides focuses on small functional peptides with robust antimicrobial activity [111]. Natural AMPs and their amino acid sequences and other properties serve as templates to design functional peptides with enhanced activity, bioavailability, and stability under physiological conditions. The synthetic AMPs must also have high selectivity and low hemolytic-cytotoxicity, and salt insensitivity [112]. A different approach is the loading of natural AMPs to a bioadhesive liquid with antibacterial activities against *S. mutans* without cytotoxicity or instability [113]. The hydrophobic content is related to the AA sequence in the nonpolar residues in the following order: Leu < Ile < Val < Thr, and < Cys [114]. The hydrophobicity index is a measure of the relative hydrophobicity, i.e., the solubility of an amino acid in the water, which is also pH dependent in the following order: Leu, Ile, Phe, Try, Val, Met, Cys, Tyr, and Ala. Both α-helical and β-sheet AMPs disrupt bacterial membranes with different specificities. In general, the α-helix peptides have more potent activity, whereas the activity of β-sheet peptides is correlated to their amphipathicity. Such parameters provide pertinent guidance for the design of sAMPs or modified natural AMPs as therapeutic drugs. Antimicrobial peptide characteristics depend on their charge, size, AA composition and sequence, hydrophobicity, and helicity (Table 7). Over 40 sAMPs have been known against different cariogenic bacteria and some selected sAMPs and their antimicrobial properties are discussed in Table 8.

Some selected examples illustrate the detailed effects of key properties of AMPs, encompassing hydrophobicity, structure, amphipathicity, and charge on their antimicrobial activity, selectivity, and plausible cytotoxicity (Table 9).

## 5. Perspectives and Concluding Remarks

### 5.1. Natural AMPs

Antibiotics and other antimicrobial agents, as discussed previously, have been used in dental care to inhibit bacterial growth and biofilm formation [173,174,175,176,177,178]. Nonetheless, extended use of such agents often leads to several adverse effects, including drug resistance and gastrointestinal disorders [179,180]. Antimicrobial agents only provide short-term antibacterial efficacy as they are diluted by saliva and degraded by salivary proteases. Without exception, antibiotic microbial resistance (AMR) to beta-lactams, tetracycline, and cotrimoxazole has appeared in cariogenic bacteria including *S. mutans* [181]. It is of great concern because this bacterium has also developed its resistance to fluoride [182]. Fortunately, no acquired resistance to carbapenems or metronidazole has been observed with streptococcal isolates [183]. An exhaustive list of several synthetic AMPs against *S. mutans*, other oral pathogens, and fungi is available in the literature [184]. Natural AMPs isolated from bacteria, animals, and plants continue to play an important role in the search for alternative treatments to overcome AMR (Table 10). Of note is the presence of azurocidin (CAP-37, NQGRHFCGGALIHARFVMTAASCFQ), a cationic protein (Mw = 37 kDa) in the gingival crevicular fluid that might be a useful biomarker of chronic periodontitis. 

One classic example is nisin (nisin A; ITSISLCTPGCKTGALMGCNMKTATCHCSIHVSK, 3354 Da; or nisin Z; ITSISLCTPGCKTGALMGCNMKTATCN_27_CSIHVSK, 3351 Da), which is produced by *Lactococcus lactis*. These two peptides have 34 Aas with amino and carboxyl endgroups, and five internal ring structures involving disulfide bridges. Nisin Z only differs from nisin A by the substitution of asparagine (N) for histidine (H) at position 27. They have four uncommon AAs: lanthionine, methyllanthionine, didehydroalanine, and didehydroaminobutyric acid. Other variants are nisin Q, nisin U, and nisin U2. There is no significant development of resistance against this polycyclic peptide as a food preservative, perhaps due to the presence of five thioether bridges in its structure [192]. The loss of rings D (Abu_23_-S-Ala) and E (Abu_25_-Ala) impairs antimicrobial activity where Abu is 2-aminobutyric acid due to due to reduced ability to penetrate the cell membrane. In contrast, the conjugation of large amino acids into ring B (Abu_8_-Ala) had a negative impact on the antimicrobial activity. Although nisin exhibits high activity against G^+^ bacteria, it lacks sufficient activity against G^−^ bacteria. Significantly enhanced activity is attained if nisin is formulated with EDTA [193] or 1-propanol [194]. Mersacidin containing β-methyllanthionine [PFSELKEAQMDKLVGAGDMEAA] is produced by *Bacillus* spp. [195]. This polycyclic peptide displays a comparable activity to that of vancomycin against MRSA (methicillin-resistant *S. aureus*) without cross-resistance [196]. Bee and wasp venoms have three valuable peptides for therapeutics: melittin, apamin, and mastoparan (MP) [197]. MP with significant cytotoxic activities serves as a model for the design of therapeutically valuable anti-infection agents from natural compounds. MP (^1^INLKALAALAKKIL^14^-NH_2_ with a net charge of +3) modified by Ala-replacement in positions 5 and 8 by isoleucine or arginine still follows the mechanism of pristine MP to disrupt bacterial membranes. The modified toxin has a net charge of +4 when the Arg is placed in position 8. The resulting modified toxin is effective against bacteria and fungi (MIC = 3–25 μM) without hemolysis or cytotoxicity against HEK-293 cells [198]. The toxicity of apamin with 18 Aas is caused by Cys, Lys, Arg, and His [199]. This peptide (CNCKAPETALCARRCQQH-NH_2_) is very rigid because two disulfide bonds are formed between Cys^1^-Cys^11^ and Cys^3^-Cys^15^ with seven hydrogen bonds. The N-terminal of melittin, a small peptide, is hydrophobic, whereas its hydrophilic C-terminal is strongly basic. Without any disulfide bridge, melittin forms a tetramer in water but spontaneously self-integrates into cell membranes. Bee venom and melittin exhibit antibacterial activities against 51 G− and G+ bacteria, MRSA, and VRE (vancomycin-resistant *Enterococcus*) with an MIC of 6 and 800 µg/mL [200]. These three peptides are useful templates to design more effective antimicrobial agents with tolerable side effects and abiding retention in the oral cavity [201]. Thus, natural peptides with antimicrobial activities against pathogens can be modified to improve stability and lower cytotoxicity. Besides bacteria and fungi, antimicrobial therapeutic peptides can be developed for targeting viruses, helminths, and protozoa. Broad antimicrobial activities and fast action are two distinct features of natural AMPs; however, they are also susceptible to proteolysis, acid hydrolysis, and instability in physiological salts. The cleaved peptides might retain some activities, as exemplified by P-113, a fragment (residue 4−16) of His-5 with activity against oral streptococci [202]. For their medical use, peptides must have good stability, high potency, and low cytotoxicity. D-peptides have considerable therapeutic potential because they are more resistant to degradation than their L-enantiomeric counterparts. 

Various AMPs, except for colistin and polymyxin B, are designed to target G^+^ bacteria, so the development of AMPs against G^−^ bacteria deserves more attention. In this context, the substitution of a single amino acid of AMPs by Trp might improve their activities and make them less vulnerable to proteolysis [203,204,205,206]. Trp positioned at the N-terminus instead of the C-terminus is more effective [206,207]. This AA binds strongly to LPS and disrupts the cytoplasmic membrane, resulting in cell lysis. His-5 binds to *P. gingivalis* hemagglutinin B, which might play a preventative role in periodontitis [208]. His-1 can attach tooth cells to teeth with damaged soft tissue to restrict microbial invasion [209]. Histatins offer an alternative treatment to candidiasis as His-5 inhibits the fungal biofilm of fluconazole-resistant *C. albicans* [210]. Of importance is the fusion of a specific targeting peptide with AMPs provided AMPs remain effective and function independently. Clinical trial C16G2 [211] selectively targets oropathogenic pathogens with significant antibacterial activity. Natural peptides with antimicrobial activities against pathogens can be modified to improve the stability and lower cytotoxicity. Besides bacteria and fungi, antimicrobial therapeutic peptides can be developed for targeting viruses, helminths, and protozoa. 

### 5.2. Antimicrobial Mechanisms of Action

The antimicrobial mechanism of AMPs deserves a concluding remark considering most AMPs are cationic with an alpha-helix structure and display electrostatic interactions with bacterial membranes with opposite charges. Polymyxins (10 Aas and 6 AA are L-α,γ-diaminobutyric acid) bind LPS in the outer bacterial membrane, leading to the penetration and disruption of the outer and inner membranes of (G^−^) bacteria [212]. However, anionic peptides isolated from microorganisms also exhibit antibacterial activities [213,214]. Therefore, other physical properties of AMPs such as the size, structure, and AA sequence also play a critical role in microbial inhibition/eradication as outlined previously. Besides the effect of AMPs on the membrane integrity, future endeavors should assess whether AMPs can alter microbial metabolic pathways or inhibit the synthesis of important biomolecules such as protein, cell wall, DNA, and RNA. 

Two important issues deserve a brief discussion concerning the design of sAMP for medical applications: stability and cytotoxicity. First, it is important to shield AMPs from proteases in biological systems. Different attempts include the insertion of artificial amino acids, cyclization, modified amino and/or carboxyl terminals, non-peptidic backbones (peptidomimetics), and multimerized AMPs [215]. There is still a lesson from His-5 with its AA sequence as it inhibits proteases and clostripain (clostridiopeptidase B, *Clostridium histolytium* proteinase B), a key function in the prevention of periodontitis. Histatins can bind to copper, nickel, and other metal ions at a physiological pH. This property can be exploited to design antimicrobial drugs with minimal bacterial resistance. Two Aas, Lys-13 and Arg-22 of His-5, are also important for antifungal activity; therefore, they should not be replaced by other amino acids [216,217]. Of note is the design of specifically targeted antimicrobial peptides (STAMPs), which target only *S. mutans* and/or other pathogens without damaging the normal flora. Conceptually, AMPs can be anchored onto the tooth surface and kill microbes upon contact. However, the method encounters several technical challenges and is not feasible in dental practice [218]. The current limitations of natural and modified AMPs can be overcome by multimerization to design compounds with improved activity and biocompatibility. This subject is beyond the scope of this review but it is available in the literature [71,219]. Another important topic is the use of advanced drug delivery systems (DDSs) to allow AMPs for oral administration. Emerging technologies such as microparticle- and nanoparticle-based DDSs are appealing for formulating both natural and synthetic AMPs [220]. 

To date, both natural and synthetic AMPs target *S. mutans*, but future research should include other common cariogenic bacteria such as *S. subrings*, *L. acidophilus*, *L.rhamnosus,* and *A. naeslundii* [221]. AMPs kill bacteria capriciously; however, the selective killing of this bacterium is feasible by creating a specific competence-stimulating peptide for *S. mutans* [222,223]. The search should be extended to *C. albicans* as only a handful of AMPs show activity against this yeast infection [224,225] as the virulence of dental caries [226]. Such a review focuses on the design of optimal synthetic AMPs with improved activities and minimal cytotoxicity using natural AMPs as guided templates. Some AMPs can bind to hydroxyapatite to prevent bacterial adhesion and others promote remineralization or mineralization of teeth by invoking the binding of calcium to hydroxyapatite [226,227]. Notice also that the topic of antiviral and immunomodulatory properties of AMPs, including LL-37 and β-defensins, is discussed elsewhere [228]. Beyond the medical field, AMPs should be explored for their applications in agriculture, food, and animal husbandry to reduce the dependency on conventional antibiotics. Different synthetic peptides are designed to target bacteria or fungi as a novel class of antimicrobial materials. Future research should include the creation of synthetic AMPs with defined AA sequences, compositions, and other properties, including charge effects, to overcome the shortcomings of natural AMPs. In general, AMPs with positive charges appear to have stronger activity than their counterparts with negative charges [229]. As an example, Attacin B with a charge of +3 [QAGALTINSDGTSGAV-VKVPITGNENHKFSALGSVDLT-NQMKL] from *Hyalophora cecropia* has a lower MIC, compared to attacin E [DAHGALTLNSDGTSGAVVKVPFAGNDKNIVSAIGSVDLT-DRQKL] with a charge of −3 from *Hyalophora cecropia* for the same four tested bacteria. However, the AA sequences of these two peptides are noticeably different, as highlighted in red, which might play a role in their interaction with bacterial membranes by hydrophobic and van der Waals forces. 

### 5.3. Nanostructured AMPs

Of note is the synthesis of self-assembled polypeptide nanogels (PNGs) to selectively combat bacterial infection [230]. Six polypeptide PNGs can be fabricated by coordination-assisted self-assembly of a mannose-conjugated antimicrobial polypeptide with Zn^2+^ ions. All PNGs are nontoxic against mammalian and red blood cells but have low MIC values (1 to 8 μg/mL) against Gram-positive *S. aureus* and Gram-negative *E. coli*. PNGs are taken up by the bacterial membrane and selectively undergo structural deformation as confirmed by isothermal titration calorimetry. This concept can be adapted or modified to eradicate oral pathogens, a subject of future endeavors. Another elegant approach is the synthesis of structurally nanoengineered antimicrobial peptide polymers (SNAPPs) against two Gram-positive bacteria (*Streptococcus mutans* and *S. aureus*) and four Gram-negative bacteria (*E. coli*, *P. aeruginosa*, *K. pneumoniae*, and *A. baumannii*) [231]. Such star-shaped peptide polymer nanoparticles consist of lysine and valine residues, synthesized via NCA (*N*-carboxyanhydrides)–ROP (open ring polymerization) of α-amino acid- NCAs, is considered a facile route for the synthesis of complex macromolecular architectures with antimicrobial properties [232]. Two specific SNAPPs have an MBC value of 3.55 ± 1.20 μM and 1.80 ± 0.14 μM, respectively, against *S. mutans*. 

Nanotechnology-based delivery systems for AMPs merit a brief discussion here since their therapeutics are limited to the topical application due to their systemic toxicity and susceptibility to protease degradation. Nanomaterials, particularly metallic nanoparticle (MNP) formulations, greatly improve the activity of antimicrobial drugs by providing support and a synergistic effect against pathogens. Various nanomaterials, including gold nanoparticles (AuNPs), silver NPs, gold nanodots, and polymeric NPs, have been tailored for the delivery of AMPs such a surfactin (a powerful bacterial cyclic lipopeptide) [233], cecropin (about 31–37 Aas and are active against both Gram-positive and Gram-negative bacteria) [234], and a pro-apoptotic peptide (a new class of anticancer agents) [235]. Albeit AuNPs exhibit bactericidal effects against MDR (multidrug-resistant) Gram-negative bacteria [236], AuNPs can be easily synthesized [237] and are commonly used as carriers. Other biocompatible materials such as liposomes, dendrimers, polymeric, and carbon nanotubes can be considered in the design of AMPs with enhanced activity toward MDR pathogens. However, several drawbacks and technical challenges are encountered such as cytotoxicity, conjugation protocols, stability profiles, and shelf-life of AMPs. Carbon nanotube synthesis is expensive with poor solubility. Biodegradable liposomes are applicable for both hydrophobic and hydrophilic drugs; however, their low loading capacity and immunogenicity remain a challenge. The cost of synthesis together with its non-specificity remain major limiting factors for dendrimers. Lipids and surfactants can be used to interact with the hydrophobic moieties of AMPs. A typical example is the formulation of cyclosporin A in poly (ethylene glycol)-b-poly(D,L-lactide-co-glycolide) nanoparticles (NPs). Such a system is smaller than 100 nm in diameter with good colloidal stability in a salt solution [238]. Liposomes are used as a carrier for nisin with antimicrobial activities against *Lactococcus lactis* [239] and *L. monocytogenes* [240]. PGG nanoparticles also serve as a carrier for nisin [241] with antimicrobial properties against *L. monocytogenes* in food preservation. The review of nanotechnology-based delivery systems for AMPs is available elsewhere [242,243]. Among various carriers such as hydroxypropyl cellulose gel [244], peptidic hydrogel [245], and gelatin microspheres [246], chewing gum is used with KSL (a decapeptide) [247] and KSL-W [248] to fight against oral pathogens. Biocompatible and biodegradable as polyvinyl alcohol, poly-L-lactic acid, polyethylene glycol, and poly(lactic-co-glycolic acid), alginate, and chitosan are extensively used in the nanofabrication of nanoparticles [249,250]. Such synthetic and natural polymers are expected to be applicable as carriers for AMPs. The potential development of nanostructured AMPs was also revisited by Yang et al. [251] with numerous relevant cited papers. AMPs can be conjugated with metal nanoparticles (gold, silver, aluminum, copper, ruthenium, etc.), carbon nanotubes, molecules/biomolecules (lipids, liposomes, cyclodextrins, dendrimers, aptamers, etc.), polymeric materials, and self-assembled peptides. Of interest are Zn-Doped CuO microparticles with antimicrobial activities against antibiotic-resistant bacteria [252]. Carbon dots (CDs) with a few nanometers in diameters are emerging nanoscale materials, and their applications as novel antimicrobial agents have been reported [253]. CDs can be doped with nitrogen, boron, sulfur, etc., and conjugated with biomolecules. CDs as well as nitrogen-doped CDs act as antimicrobial agents as their surfaces encompass functional hydroxyl, carboxyl, and amino groups that generate free radicals to eradicate bacteria [253]. The surface carboxyl or amino groups can be conjugated with AMPs toward the development of nanostructured AMPs, a subject of future endeavors. 

Lastly, an emerging area of growing interest is the use of AMPs together with conventional antibiotics, in a synergistic mode of action [254]. These combined therapies appear as a promising approach due to the different modes of action of AMPs and antibiotics. Thus, the multisite antimicrobial action of AMPs and antibiotics contributes to rapid eradication to neutralize any microbial resistance. This combination is very critical as Gram-negative bacteria can synthesize 4-aminoarabinose or palmitoylation in lipid A in their membrane to reduce the negative charge, preventing their electrostatic interaction with AMPs. 

## 6. Conclusions

Native and cationic AMPs, mainly LL-37, defensins, and histatins, have antimicrobial properties against numerous oral pathogens including *S. mutans* and *C. albicans* without antimicrobial resistance. Their susceptibility to proteolysis and short half-lives can be improved by chemical modification to overcome these limitations. The modified AMPs should also possess remineralizing properties for hydroxyapatite to prevent and treat caries. However, more investigations are needed to design optimal AMPs with high activities and minimal cytotoxicity. AMPs must also be designed and formulated for oral administration. Synthetic AMPs can be used alongside conventional antibiotics to combat dysbiosis, an “imbalance” in the gut microbial community, and prevent oral infections. In this context, synthetic AMPs should also be designed to reshape the oral microbial community and their use in synergy with conventional antibiotics is promising to treat multi-drug resistant bacterial infections. The low success rate of AMPs for clinical applications must be addressed, albeit numerous natural new peptides have been isolated, identified, and modified continually. Research activities for AMP-nanocarrier optimization are underway to identify AMP-carrier systems with high entrapment efficiency and facile conjugation chemistry. AMPs are also subject to chemical modifications and/or novel formulations to be less toxic, more bioavailable, and useful in biomedical applications. A combination of nanostructured AMPs and conventional antibiotics as powerful antimicrobial agents is awaiting exploitation.

## Figures and Tables

**Figure 1 jfb-13-00175-f001:**
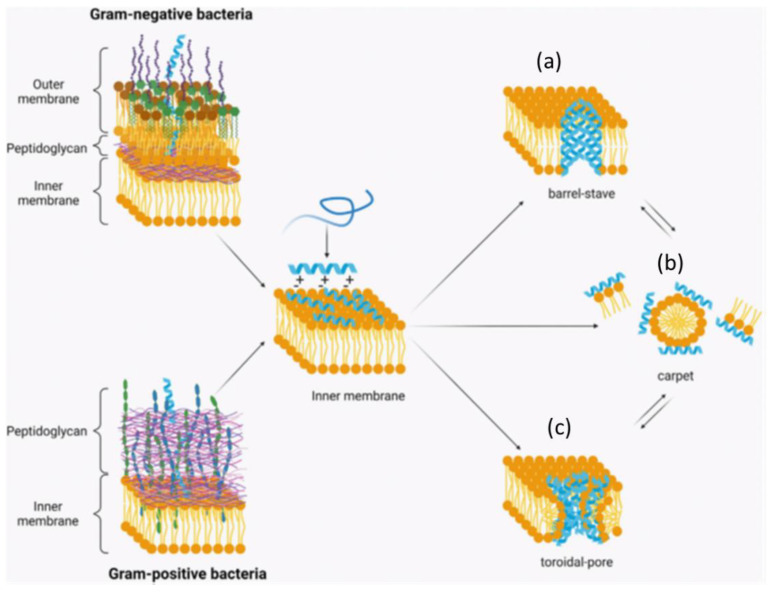
Electrostatic interactions between cationic peptides with amphophilic/amphipathic properties and negatively charged cell membranes [70]. (**a**) Barrel-stave model, (**b**) carpet model and (**c**) toroidal model [71]. The hydrophilic region effects the spot-on peptide alignment on the bacterial membrane [71]. The figure is reproduced from Ref. [71] with open access.

**Table 1 jfb-13-00175-t001:** The use of small compounds in oral therapeutics.

Small Molecules	General Observations	Ref.
Fluoride (F^−^, an ionized form of the element fluorine, F)	A key component in potable water, mouthwashes, toothpaste, and oral supplements to ward off dental cavities	
	Binding to the tooth surface to promote remineralization and balance acid-stimulated demineralization	[24]
	Fluoride impedes the enolase activity of *S. mutans* and other *Streptococci*. The enzyme reversibly catalyzes D-2-phosphoglycerate and phosphoenolpyruvate in glycolysis and gluconeogenesis.	[27,28]
	Fluoride might invoke dental and skeletal fluorosis and emerging fluoride-resistant oral bacteria.	[29]
Chlorhexidine- a cationic polybiguanide-C_22_H_30_Cl_2_N_10_, Mw = 505.45 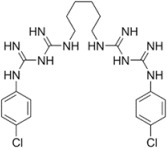	A “gold standard” of antiplaque agents.	[30]
	Effective against bacteria and yeasts by disrupting their inner cell membranes.	[30,31]
	Plaque adhesion is prevented as the acidic groups of salivary glycoproteins are blocked by chlorhexidine.	
	Reduces the bacterial attachment to tooth surfaces by competing with Ca agglutination.	[32]
	Inducing DNA damage in oral mucosal cells, white blood cells, kidney cells, and cellular apoptosis.	[33,34]
Quaternary Ammonium Salts (positive charge)	Widely used in mouth rinses to inhibit oral plaque.	[35]
	Binding to the negatively charged bacterial cells to invoke bacterial lysis.	[36]
	Several side effects: convulsions, hypotension, gastrointestinal symptoms, coma, and even fatality.	[37]
Triclosan- C_12_H_7_Cl_3_O_2_, Mw = 289.54 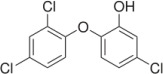	Added to many consumer products including toothpaste to reduce or prevent bacterial contamination.	[38]
	This broad-spectrum agent blocks enoyl–acyl carrier protein reductase, an enzyme required for microbial lipid synthesis	[39]
	Susceptible to bacterial resistance.	[40]
	By 2015, Johnson & Johnson had removed triclosan from all of its products.	[41]

**Table 2 jfb-13-00175-t002:** Systemic antibiotics and their uses in the treatment of oral diseases.

Antibiotics	Mechanisms, Drug Resistance, and Side Effect	Ref.
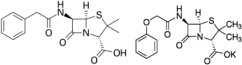 Penicillin: β-lactam antibiotic Penicillin G (benzylpenicillin) or penicillin V (phenoxymethylpenicillin) is frequently prescribedPenG- C_16_H_18_N_2_O_4_S, Pen VMw = 334.4 C_16_H_18_N_2_O_5_SMw = 350.39	Inhibits the formation of peptidoglycan in the cell walls.	
Effective against G^+^ *Streptococci* and *Staphylococci*, and some G^−^ bacteria.	[49]
Bacterial resistance as they produce mecA that encodes PBP2a with low binding affinity to β-lactams.	[50,51]
Diarrhea, nausea, rash, urticaria, hypersensitivity, and neurotoxicity.	[52]
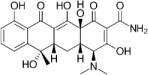 Tetracyclines-C_22_H_24_N_2_O_8_, Mw = 444.440	Binds to the mRNA translation complex via its 30S ribosomal subunit to inhibit protein synthesis.	[53,54]
Cramps, stomach burning, diarrhea, sore mouth, nephrotoxicity, non-oliguric acute renal failure, and teeth discoloration.	[55,56,57]
Metronidazole- C_6_H_9_N_3_O_3_, Mw = 171.16 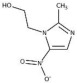	Effective against oral obligate anaerobes as it inhibits nucleic acid synthesis by disrupting DNA.	[58]
Causing several side effects including nausea, headaches, and tachycardia.	[59]
Potential use in treating periodontitis	
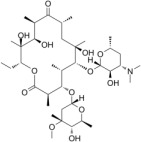 Macrolides (Erythromycin: C_37_H_67_NO_13_,MW = 733.93)	Effective against *Streptococci*, *Staphylococci*, *Pneumococci*, and *Enterococci.*	[60]
Myopathy, enterohepatic recycling, and cholestasis.	[61]
Erythromycin can decrease 35% of the plaque amount after one week.	[62]
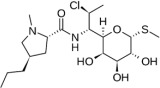 Clindamycin C_18_H_33_ClN_2_O_5_S,Mw = 424.98	Inhibits bacterial protein synthesis by disrupting ribosomal translocation.	[63]
Effective against anaerobic bacteria.	[64]
An alternative to treat patients with allergy to penicillin or penicillin-resistant infection.	
Diarrhea, nausea, abdominal pain, vomiting pseudomembranous colitis, and contact dermatitis	[65]

**Table 3 jfb-13-00175-t003:** Classification of amino acids (AAs).

Amino Acids	Abbreviation with 3 Letter Code or One Single Capital Letter Code
Polar (6 AAs) but not charged	Asparagine (Asn, **N**), Cysteine (Cys, **C**), Glutamine (Gln, **Q**), Threonine (Thr, **T**), Tyrosine (Tyr, **Y**), Serine (Ser, **S**)
Hydrophobic (9 AAs)	Alanine (Ala, **A**), Glycine (Gly, **G**), Isoleucine (Ile, **I**), Leucine (Leu, **L**), Methionine (Met, **M**), Phenylalanine (Phe, **F**), Proline (Pro, **P**), Tryptophan (Trp, **W**), Valine (Val, **V**). Mostly their carbon and hydrogen, have very small dipole moments and tend to be repelled from water
Negative charged (pH 7) (2 AAs)	Aspartate (Asn, **D**), Glutamate (Glu, **E**)
Positive charges (pH 7) (3, AAs)	Arginine (Arg, **R**), Histidine (His, **H**), Lysine (Lys, **K**),

**Table 4 jfb-13-00175-t004:** The sequence of amino acids of various histatins.

Histatins	General Observations, AA Sequence of Histatins (One Letter Code)
His-1 (salivary glands-bone marrow): 38 AA, Mw ∼4929 Da	DSPHEKRHHGYR [His-2 or RKFHEKHHSHREFPFYGDYGSNYLYDN]
His-5 (salivary glands): 24 AA, Mw ∼3037 Da	DSHA[His-12]R[His-8] or DSHA-KRHHGYK- R-KFHEKHHSHRGY
His- 3: 32 AA, Mw ∼4063 Da Proteolytic fragments in saliva	[His-5]RSNYLYDN or [His-6]SNYLYDN
His-2 (salivary glands) (27 AA)	RKFHEKHHSHREFPFYGDYGSNYLYDN
His-4 (20 AA)	[His-7]RSNYLYDN or RKFHEKHHSHRGY-RSNYLYDN
His-6 (25 AA)	[His-5]-R
His-7 (13 AA)	R[His-8] or R-KFHEKHHSHRGY
His-8 (12 AA)	KFHEKHHSHRGY
His-9 (14 AA)	[His-7]R or RKFHEKHHSHRGY-R
His-10 (13 AA)	[His-8]R or KFHEKHHSHRGY-R
His-11 (8 AA)	[His-12]R or KRHHGYK-R
His-12 (7 AA)	KRHHGYK

**Table 5 jfb-13-00175-t005:** Antimicrobial activities of hBD against some selected oral bacteria.

hBD	Microorganisms and Minimum Inhibition Concentrations (MUC)	Ref.
hBD1,2,3	- Activities against *S. mutans*, *E. faecalis*, and other oral pathogens. hBD-2 and -3 are inducible by the bacterial LPS, whereas hBD-1 averts normal flora from becoming opportunistic.	[93]
hBD-2	- Effective against *C. albicans* and is also induced by lichen-planus related inflammation.	[94]
β-defensins	- *Aggregatibacter actinomycetemcomitans* and *Fusobacterium nucleatum* are two periodontal pathogens.	[95]
hBD-3	- MIC = 12.5 mg/L for *F. nucleatum*, 100–200 mg/L for *P. gingivalis*, *A. actinomycetemcomitans*, and *Prevotella intermedia*.	[96]
hBD-2/hBD-3	- MIC = 3.9–250 μg/mL and 1.4–250 μg/mL) for several oral pathogens and *Candida* spp.	[97]
*β*-defensins	- Pathogens including *T. denticola* likely interfere with the signal pathway to subdue the expression of *β*-defensins to develop the resistance against the peptides	[98]

**Table 6 jfb-13-00175-t006:** Type, expression site, and AA sequence of defensins.

α-Defensins	Sequence One Letter Code	
hNP-1 (30 AAs)	A-[hNP-2] (Disulfide bridge (DB): 2-30, 4-19, 9-29)	
hNP-2 (29 AAs)	CYCRIPACIAGERRYGTCIYQGRLWAFCC Begins and ends with cysteine residues. DB: 1-29, 3-18, 8-28	
hNP-3 (30 AAs)	D-[hNP-2] (DB: same as hNP-1)	
hNP-4	VCSC RLVFC RRTEL RVGNC LIGG VSFTY CCTRVD (DB: same as hNP-1)	
α-defensins regulate complement activation and enhance macrophage phagocytosis.		[101]
**β-defensins**	Sequence One Letter Code	
hBD-1	DHYNCVSSGG QCLYSACPIF TKIQGTCYRG KAKCCK (36 AA residues with six Cys forming three intramolecular disulfides) *P. gingivalis*, *A. actinomycetemcomitans*, and *F. nucleatum*	
hBD-2	GIGDPVTCLK SGAICHPVFC PRRYKQIGTC GLPGTKCCKK P Effective against G-bacteria and *C. albicans*, but less effective against G^+^ bacteria.	[102]
hBD-3	GIINTLQKYY CRVRGGRCAV LSCLPKEEQI GKCSTRGRKC CRRKK (45 AAs, Disulfide bridge: 11-40, 18-33, 23-41) Effective against several bacteria including *S. mutans* and *P. gingivalis*	

**Table 7 jfb-13-00175-t007:** Physicochemical properties of AMPs and their antimicrobial properties.

Charge	Ref.
A charge of +1 provided by Arg or Lys residues is needed but there is no correlation between high positive charges with high activities.	[115]
However, high positive charges (>+9) are related to high hemolytic activities. The Lys substitution is less hemolytic than the Arg	[116]
substitution. In general, analogs with net positive charges invoke high cytotoxicity.	
**Hydrophobicity/Hydrophobic Moment (HM)**	
The second strongest force behind the charge drives AMPs (to the bacterial membrane). A single AA replacement might alter hydrophobicity and subsequently activity. The α-helical peptide with high hydrophobicity of its nonpolar face possesses high activities.	[117,118,119]
However, there is limited hydrophobicity to attain maximal activity without any undesirable effects. The side chain and bulkiness of AMPs also affect the antimicrobial activity. Phe > Ile > Leu > Val for the order of side chain length and Phe > Leu > Ile > Val for bulkiness. Increasing overall hydrophobicity might increase activity and hemolysis, provided there is no change in amphipathicity, helicity, or net charge. The hydrophobic moment (HM) or an index of amphipathicity is also attributed to antimicrobial activity, indicating the peptide’s capability to switch from a polar face to a nonpolar face right after its insertion into the membrane.	[120,121]
The hydrophobic moment is defined as the vertical sum of individual AA hydrophobicity of a specific peptide over that of an ideal α-helix peptide. For instance, Hp1404-T1 (HM = 0.699), has 16-fold less antimicrobial activity compared to Hp1404-T1e with an HM of 0.831.	[121]
Compared to hydrophobicity, helical amphipathicity is a stronger force in interfacial binding.	[122]
**Length**	
AMPs often have <20 amino acid residues (AAR).	[123]
A peptide transversing the lipid bilayer requires 18 AAR, 2–4 AAR per turn for an α-helix, and 7–8 AAR (two turns) for amphipathic faces.	[124]
The smaller peptides are considered safer than their bigger counterparts.	[116]
Compared to melittin (15 AA), an analog derived from the peptide C-terminus exhibits 7-fold lower antimicrobial activity and 300-fold lower hemolytic activity.	[125]
**AA Sequence**	
A proper peptide sequence often results in an α-helix and histidine is not often found in AMPs, as its size impairs the AMP insertion into the membrane.	[126,127,128]
Arg is most critical to antimicrobial activity to display both hydrophobic and electrostatic interactions with bacterial membranes.	[127,128]
The cysteine function in AMPs is still unknown; however, the replacement of Cys with Ala or Leu reduces the hBD-1 activity. Peptides could not penetrate the membrane if their cysteines form disulfide bridges.	[129]
**Self-Association**	
To a certain extent, peptide self-association (dimerize) enhances antimicrobial activity	[130]
However, strong self-association prevents the penetration of AMPs into the cytoplasmic membrane.	

**Table 8 jfb-13-00175-t008:** The design and performances of synthetic AMPs.

Synthetic AMPs: Design and Performances
AMPCol on smooth titanium surface: A synthetic antimicrobial peptide (Tet213, KRWWKWWRRC) is conjugated to free amines of collagen IV to form AMPCol. The coating of AMPCol on smooth titanium surfaces sustains its release to prevent peri-implantitis [131]. C16G2: It targets *S. mutans* and *S. salivarius*. It shows an overall potency of C16G2 against G- species. C16G2 is also effective against two G+ bacteria from human skin flora [132]. Peptide (CSP), sequence (SGSLSTFFRLFNRSFTQALGK), CSPC16, preserves pheromone activity, whereas the 8-TFFRLFNR region within CSPC16, CSP M8, is sufficient for its specific delivery to *S. mutans* [133]. D1–23 (FLPKTLRKFFARI RGGRAAVLNA) with low cytotoxicity outperforms chlorohexidine against *S. mutans* biofilm and it is also effective against *S. mitis* and *S. salivarius* [134]. It also shows D-GL13K (GKIIKLKASLKLL-NH_2_) [135]. PR39 (RRRPRPPYLP RPRPPPFFPP RLPPRIPPGF PPRFPPRFP-NH_2_ (only 7 different AAs) [136] and VSL2 [AcA∆FKA∆FWK∆FVK∆FVK-NH_2_] where Ac is an acetyl group [137] and ∆F = ΔPhe, α,β-dihydrophenylalanine, show activity against *E. faecalis* at different levels [138]. DJK-5: VQWRAIRVRVRVIR shows activities against *E. faecalis*, *P. aeruginosa,* and a mixture of *S. aureus*, *S. epidermidis* [139]. GH12: α- helical AMP with a sequence of GLLWHLLHHLLH-NH_2_ is effective against oral *Streptococci* and reduces the exopolysaccharide production by *S. mutans* [140,141]. HBD3–15: Derived from hBD-3, consists of 15 AAs (GKCSTRGRKCCRRKK). Effective against *S. mutans*, *S. gordonii*, and *E. faecalis* on biofilm formation [134,142]. IDR-1002 with a sequence of VQRWLIVWRIRK-NH_2_ exhibits antibacterial properties and anti-inflammatory of *S. aureus* and *E. faecalis* [143]. KSL [KKVVFKVKFK-NH_2_] KSL inhibits oral pathogens associated with dental caries and plaque [144]. KSL is cleaved at K6–V7 in the human saliva and F5–K6 in simulated gastric fluids [145]. An analog of KSL, where the L6 residue is replaced with W, is stable and preserves its activity against several oral pathogens [146]. P-113/Nal-P-113: Biocompatible His-5 is easily degradable in saliva, plasma, and serum [147,148]. Nal-P-113 is synthesized from His-5 (AKRHHGYKRKFH) by substituting histidine with β-naphthylalanine. This modified peptide is highly stable in physiological conditions and is effective against *P. gingivalis*, an asaccharolytic G^-^ anaerobic bacterium. ToAP2 (FFGTLFKLGSKLIPGVMKLFSKKKER, 26 AA) and NDBP-5.7 (ILSAIWSGIKSLF-NH_2_), the two peptides are effective against *C. albicans*, but the former is more active and effective than the latter [149].

**Table 9 jfb-13-00175-t009:** Selected AMPs with antimicrobial activities, selectivity, and cytotoxicity.

Activity/Selectivity/Cytotoxicity
**Charge effect** AR-23 (melittin-related peptide with 23 AA, a charge of +4, and an α-helical amphipathic structure, GIGAVLKVLTTGLPALISWIKRKRQQ-NH_2_). If Ala is replaced by Arg or Lys, a variant with 3 substitutions acquires a net charge of +7 with reduced hemolytic activity. Hemolysis is more pronounced with the Arg substitution than the Lys substitution. Interpretation of the charge effect is complicated and nebulous as the substitutions also alter the amphipathicity, helicity, and hydrophobicity of the peptide [150]. Aurein 1,2 (53.04% α-helix) is a natural peptide with 13-AA (GLFDIIKKIAESF) and a net charge of +1. Replacement of Asp 4 and Glu 11 with Lys increases the net charge to +5. The modified peptide (78.44% α-helix) exhibits improved efficacy against *E. coli* and *P. aeruginosa* [151]. Further substitution, Ala 10 for Try, the resulting peptide with 36.36% α-helical content exhibits higher antimicrobial activity and lowest hemolysis. Such results are more relevant to the charge increase, not the increased α-helicity. The AA sequence and the Trp substitution position are also two major contributing factors. V13Kʟ with 26 AA and a net charge of +7: [Ac-AKWKSFLKTFKSAKKTVLHTALKAISS–amide] On the polar face, a variant of this peptide with a charge decrease to +4 significantly reduces both antimicrobial and hemolytic activities. A variant (+8) acquires higher antimicrobial activity, whereas the hemolytic activity remains unchanged. Any further increase in charge improves activity, however, the peptide becomes significantly more hemolytic [152].
**Hydrophobicity vs. hemolysis** V13Kʟ: If Val 16 is substituted with Leu, the variant (V16L) becomes more hydrophobic with slightly increased activity. However, it causes 53.9% hemolysis vs. 28.3% [153]. When Val 16 is substituted with Ala, a new variant (V16A) becomes less hydrophobic and causes only 14.3% hemolysis. It has the same MIC for clinically isolated *P. aeruginosa* ATC2853 and *E. coli*, compared to V13Kʟ. C18G: A platelet factor IV-derived AMP [ALWKKLLKKLLKSAKKLG] The substitution of Leu to Phe or Ile has no effect on its MIC against 5 tested bacteria. The replacement by Val, however, increases the MIC values against all such bacteria. The effect is more pronounced with α-aminoisobutyric acid substitution [154].
**Amino acid sequence and composition** Lys and Arg with similar charges are often used to design synthetic AMPs. Higher activity is obtained with the Arg substitution over the Lys substitution [155]. The Arg side chain interacts with two bacterial lipid head groups, compared to one for the Lys side chain [156]. Arg-rich AMPs by ion-pair–π interactions promote enhanced peptide-membrane interactions as exemplified by indolicidin [ILPWKWPWWPWRR-NH_2_] and tritrpticin [VRRFPWWWPFLRR], two Arg and Trp rich peptides [157]. Octa 2 with a sequence of RRWWRWWR is another Trp- and Arg-rich peptide with activities against *E. coli*, *P. aeruginosa,* and *S. aureus*. Short Trp- and Arg-rich AMPs can be derived from murine or bovine lactoferricin with strong activities [158]. D-amino acids might have higher activities than ʟ-amino acids. The D-form of sapecin B (88 AAs) has a significantly lower MIC than its L-isomer counterpart: 16-fold with *S. aureus* and only 2-fold with *E. coli* [159]. However, the D-forms and L-forms of Kn2–7 [FIKRIARLLRKIF], mastoparan M [INKAIAALAKKLL-NH_2_], and temporins (10–14 Aas) have comparable activities. Phen 3 and 13 in Aurein 1.2 bind to bacterial membranes, thus their substitution by Ala decreases activity [160]. Proline (nonpolar AA)-rich peptides (PrAMPs) enter the cytoplasm via the inner membrane transporter SbmA to target ribosomes or interfere with protein synthesis [161]. Octa 2: replacing Arg residues with His, the modified peptide shows good therapeutic potential [162]. L4H4 [NH_2_-KKALLAHALAHLALLALHLALHLKKA-Amide]: by inserting four His sequences in Leu and Ala, the modified amphiphilic magainin shows good properties for cell penetration and antibacterial activity [163]. Attacins and diptericins have 14% to 22% glycine residues [164,165]. Salmonid cathelicidins (glycine-rich) activate phagocyte-mediated microbicidal activity, unlike the conventional mode of AMPs [166]. The glycine-rich central–symmetrical GG3 is considered a candidate against G-bacteria [167]. Cys-rich peptides (defensins). Antimicrobial activities stem from their unique amino acid sequence against G^−^ and G^+^ bacteria, fungi, and enveloped viruses.
**Side change** The N-terminal with added Acyl groups or fatty acid chains enhances the antimicrobial activity; in contrast, the use of octanoyl, or 6-methyl-octanoyl results in weak activity [168]. LL-37 [LLGDFFRKSKEKIGKEFKRIVQRIKDFLRNLVPRTES] with increasing hydrophobicity, and activity is obtained when different-length fatty acid chains are added to the peptide. Increasing alkyl chain lengths beyond 8 carbons (C8), however, reduces activity as the modified AMP is more prone to self-assembly [169]. C8-KR12-NH_2_ shows only <10% hemolysis at very high concentrations. However, the incorporation of C10 significantly increases hemolytic activity. Adding fatty acid chains to anoplin (an amphipathic, α-helical peptide of wasp venom, [GLLKRIKTLL-NH_2_] results in improved activity with increased hemolysis. The latter is correlated to the chain length though hemolysis is only 10% [170]. The lipidation effect on AMP activity is further reviewed elsewhere [171,172].

**Table 10 jfb-13-00175-t010:** Some selected natural AMPs from different sources.

Natural AMPs	Characteristics and Antimicrobial Activities	Ref.
Amphibian-Derived	Magainin 1 [GIGKFLHSAGKFGKAFVGEIMKS] and magainin 2 [GIGKFLHSAKKFGKAFVGEIMNS], skin secretions of frogs. Both have 23 Aas each and differ by two substitutions.	[185]
Insect-Derived	Cancrin [GSAQPYKQLHKVVNWDPYG] (The first peptide from the sea amphibian *Rana cancrivora*). Cecropin-31–37 Aas (guppy silkworm, bees, *Drosophila*).	[186]
Microorganisms-Derived	Nisin and gramicidin (linear peptides with 15 amino acids) from *Lactococcus lactis Bacillus brevis*, and *Bacillus subtilis*.	[187]
Plant-derived	Thionins (45–48 AA, 6–8 Cys forming 3–4 disulfide bonds). The C-terminal region with 12 conserved Cys residues contributes to its stability).	[188]
Marine-derived	As-CATH4 (immunity-stimulating effect in vivo). Myticusin-beta is a promising antimicrobial agent. The GE33-based vaccine can enhance antitumor immunity in mice. GE33 is also known as pardaxin with the following sequence: GFFALIPKISSPLFKTLSAVGSALSSSGGQE-OH.	[189,190,191]

## Data Availability

Not applicable.

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
