# Peer review of "Important Roles and Potential Uses of Natural and Synthetic Antimicrobial Peptides (AMPs) in Oral Diseases: Cavity, Periodontal Disease, and Thrush"

_jfb, 2022, doi:10.3390/jfb13040175_

Round 1
Reviewer 1 Report
In this submission, the author has summarized the strategy of the design and synthesis of antimicrobial peptides (AMPs), their antibacterial activity and blood cell toxicity. Besides, the review also highlights natural antimicrobial peptides. It’s fact that synthetic cationic peptides are hemolytic which precludes their wide application in vivo. Certainly, along with peptides, higher molecular weight polypeptide also faces an identical issue. So, a brief section on how nano or micro formulation strategy can reduce toxicity by keeping the same level of antibacterial activity of cationic peptide/polypeptide would be a worthwhile addition to this review. None other less, the manuscript has been well written and could be published after considering the following comments. Here are my comments which need to be addressed.
1. It would be good to add a brief discussion on how nano formulation helps in reducing the toxicity of antimicrobial peptides/ polypeptides (as mentioned above).
The author can refer to these key papers.
a. Coordination-Assisted Self-Assembled Polypeptide Nanogels to Selectively Combat Bacterial Infection.
b. Combating Multidrug-resistant Gram-negative Bacteria with Structurally Nanoengineered Antimicrobial Peptide Polymers.
2. Table 7:
'-The hydrophobic moment is defined as the vertical sum of individual AA hydrophobicity of a specific peptide over that of an ideal α-helix peptide.
- For instance, Hp1404-T1 (HM = 0.699), has16-fold less antimicrobial activity compared to Hp1404-T1e with an HM of 0.831'
These two sections could be one section, as they saying the same context.
3. In table 8, in some place colon symbol, is missing. Please make it consistent.
4. Reference number ‘56]’ needs to be fixed without the bracket. Also, please double-check with the journal name of reference 57.
Author Response
The rebuttal to Reviewer #1 is attached

Reviewer 2 Report
According to the authors, the review should focus on the possibility of using natural and synthetic antimicrobial peptides to treat oral diseases. Unfortunately, this task has not been accomplished, as the review is limited only to AMPs for the treatment of caries. Other viral or bacterial diseases of the oral cavity, including oncological, are out of the question. However, the presentation of data on AMPs produced by oral tissue cells, their unique biological effects and their use in clinical practice seems to be of interest in this aspect.
Therefore, the material contained in tables 1 to 2 seems redundant and unnecessary for review purposes. Table 3 - Repetition of common places, which is clearly excessive.
There is no explicit focus on the biocidal action of AMPs directed at bacteria causing oral diseases.
Some minor remarks:
Table 9 (p. 12) shows the incorrect structure of the peptide V13Kl, which clearly exceeds 26 Aa and has a different charge.
In addition, MSTKDFNLDLVSKKVSDSGASPR (p.14) is a leader peptide (23 Aa), Nisin Z has a ITSISLCTPGCKTGALMGCNMATCNCSIHVSK sequence. It should be noted that the activity of the nisin is not due to the presence of uncommon amino acids, but the multiple (5) thioether bridges which they form.
If you carefully compare the Attacin B and E peptide sequences (p.15-16)
QA GALT I NSDGTSGAVKVP IT GN ENHKF SA L GSVDLT NQM KL
DAH GALT L NSDGTSGAVKVP FA GN DKNIV SA I GSVDLT DRQ KL
It becomes obvious that they differ not only in charge, but also in sequence.
In connection with the above, it is necessary to carry out a thorough check of the listed structures of peptides.
Author Response
The rebuttal to Reviewer #2 is attached

Round 2
Reviewer 2 Report
The latest version of the review contains many revisions and additions, and seemed quite worth publishing. Two remarks - the structure of the leader peptide rather than the primary structure of nisin is still shown on p.16. And the insert at the same paragraph looks much more correct in the file containing answers to previous comments.
